# Deep reptilian evolutionary roots of a major avian respiratory adaptation

Yan-yin Wang [1✉], Leon P. A. M. Claessens [2] & Corwin Sullivan[1,3]

Vertebral ribs of the anterior thorax in extant birds bear bony prongs called uncinate processes, which improve the mechanical advantage of mm. appendicocostales to move air through the immobile lung and pneumatic air sacs. Among non-avian archosaurs, broad, cartilaginous uncinate processes are present in extant crocodylians, and likely have a ventilatory function. Preserved ossified or calcified uncinate processes are known in several non-avian dinosaurs. However, whether other fossil archosaurs possessed cartilaginous uncinate processes has been unclear. Here, we establish osteological correlates for uncinate attachment to vertebral ribs in extant archosaurs, with which we inferred the presence of uncinate processes in at least 19 fossil archosaur taxa. An ancestral state reconstruction based on the infer distribution suggests that cartilaginous uncinate processes were plesiomorphically present in Dinosauria and arguably in Archosauria, indicating that uncinate processes, and presumably their ventilatory function, have a deep evolutionary history extending back well beyond the origin of birds.

[1] Department of Biological Sciences, CW 405 Biological Sciences Building, University of Alberta, Edmonton, AB T6G 2E9, Canada. [2] Maastricht Science Programme, Faculty of Science and Engineering, Maastricht University, Maastricht, The Netherlands. [3] Philip J. Currie Dinosaur Museum, Wembley, AB T0H 3S0, Canada. ✉email: yanyin@ualberta.ca

Extant birds and crocodylians are modern representatives of Archosaurs, a group of amniotes that first appeared in the Triassic and filled most niches available to large-bodied terrestrial vertebrates throughout the Jurassic and Cretaceous[1,2]. The anterior thoracic vertebral ribs of almost all extant archosaurs bear posteriorly protruding uncinate processes, although these structures are typically lacking in anhimid and megapodid birds[3]. Some neognath birds (e.g. *Falco sparverius* (UAMZ 4022)) have additional uncinate processes on the posteriormost cervical ribs. In most extant birds, uncinate processes take the form of slender, bony protrusions that are typically fully ossified[4,5] and fused to the vertebral ribs[6] in skeletally mature individuals (Fig. 1a–d), whereas in extant crocodylians the uncinate processes take the form of cartilaginous tabs (Fig. 1e, f)[7–9]. The rod-like cartilaginous uncinate processes of the rhynchocephalian *Sphenodon punctatus* may not be homologous to those of birds and crocodylians: most extant lepidosauromorphs lack uncinate processes[10], and whether uncinate processes were ancestrally present is currently ambiguous for both Lepidosauromorpha and Sauria. Two main hypotheses for the function of avian uncinate processes, which are not mutually exclusive, have been proposed: mechanical reinforcement of the ribcage[11–14] and ventilation[15,16]. The ventilatory hypothesis is bolstered by theoretical analyses suggesting that uncinate processes should act to increase the leverage of the muscles attached to them (e.g. mm. appendicocostales), which in turn should enhance the bird's ventilatory

performance as the expansions/contractions of the thorax move air through the respiratory organs[15,16]. The mechanical leverage provided by uncinate processes likely facilitates the ventilatory motions of the ribcage measured in in vivo studies[17,18]. Furthermore, the ventilatory hypothesis has received experimental support, from an electromyographic study carried out on mm. appendicocostales in the Canada goose *Branta canadensis*[19], and uncinate processes accordingly represent an important component of the highly specialized avian ventilatory system, along with pneumatic sacs, unidirectional pulmonary airflow, and cross-current gas exchange[14,20,21]. In extant crocodylians, unidirectional pulmonary airflow is also present[22,23], and m. iliocostalis, an epaxial muscle attached to the uncinate processes, has been found empirically to serve an expiratory function in the American alligator *Alligator mississippiensis*[24]. Though the specifics of function and homologies involving the respiratory organs and the process of gas exchange are not yet fully understood[25], an anatomical capacity for unidirectional pulmonary airflow and a ventilatory mechanism incorporating uncinate processes may have been ancestral for Archosauria.

The fossil record has provided clear evidence for uncinate processes in a small number of archosaurs outside the extant avian and crocodylian crown groups. Slender, ossified uncinate processes have been reported in some non-avian members of Pennaraptora[6], the clade of theropod dinosaurs containing birds and their closest relatives. Ossified uncinate processes are also

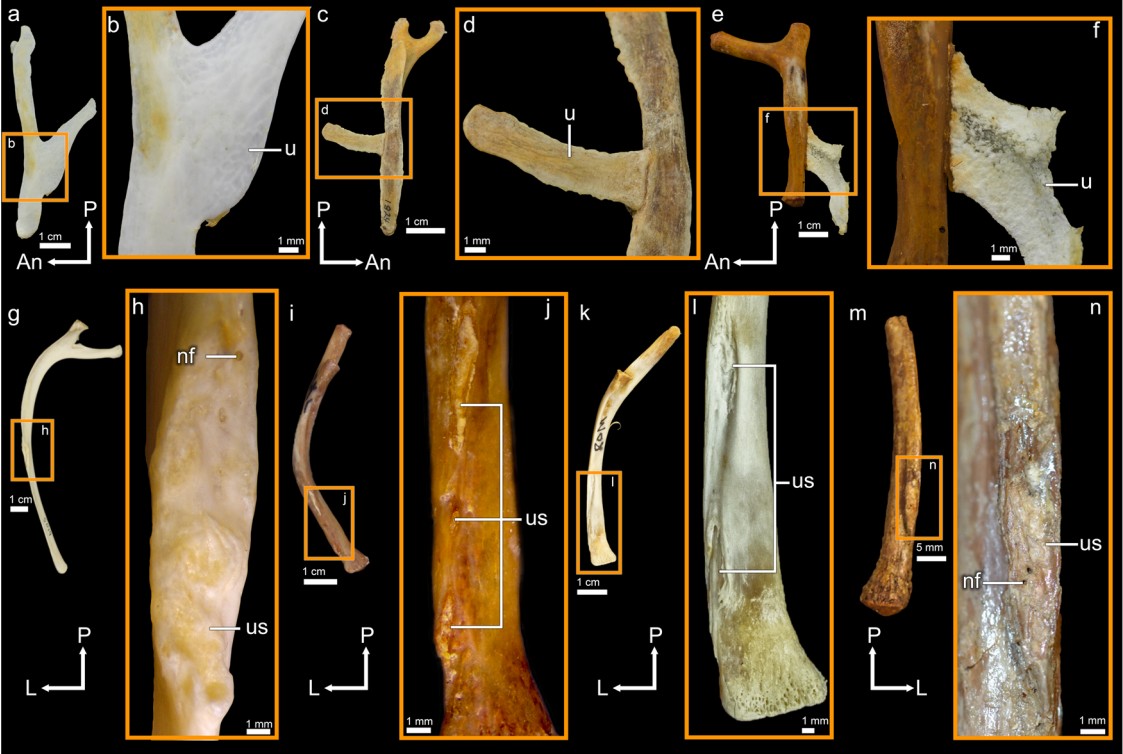

**Fig. 1 Dorsal vertebral ribs with uncinate processes and uncinate scars in extant archosaurs, and vertebral rib with uncinate scar in an indeterminate fossil crocodylian.** Left dorsal vertebral rib of the bald eagle *Haliaeetus leucocephalus* (UAMZ 5028) in lateral view (**a**), and close-up of contact between dorsal vertebral rib and uncinate process in lateral view (**b**); right dorsal vertebral rib of the skeletally immature wild turkey *Meleagris gallopavo* (UAMZ 1824) in lateral view (**c**), and close-up of contact between dorsal vertebral rib and uncinate process in lateral view (**d**); left dorsal vertebral rib of the alligatorid crocodylian *Caiman crocodilus* (UAMZ unnumbered) in lateral view (**e**), and close-up of contact between dorsal vertebral rib and cartilaginous uncinate process in lateral view (**f**); left dorsal vertebral rib of the palaeognath bird *Rhea americana* (UAMZ 5019) in posteromedial view (**g**), and close-up of uncinate scar in posterior view (**h**); left dorsal vertebral rib of *C. crocodilus* (UAMZ unnumbered) in posteromedial view (**i**), and close-up of uncinate scar in posterior view (**j**); left dorsal vertebral rib of *C. crocodilus* (ROM R7077) in posteromedial view (**k**), and close-up of uncinate scar in posterior view (**l**); incomplete right vertebral rib of indeterminate fossil crocodylian from the Miocene of Florida, USA (AMNH 7900) in posteromedial view (**m**), and close-up of uncinate scar in posterior view (**n**). Branching lines indicate multiple rugose areas of a single uncinate scar separated by smooth cortex. An anterior, L lateral, nf nutrient foramen, P proximal, u uncinate process, us uncinate scar.

known in the ornithomimosaurian theropod *Pelecanimimus polyodon*[26]. Moreover, triangular calcified uncinate processes are present in the notosuchian *Araripesuchus gomesii*[27], and broad calcified uncinate processes termed intercostal plates have been observed in several ornithischian dinosaurs including ankylosaurs, thescelosaurids, ornithopods, and stegosaurs[28–31]. However, the notosuchian and ornithischian examples listed above notwithstanding, preserved uncinate processes are rarely found outside Pennaraptora. This could indicate that uncinate processes are unusual structures that evolved independently in crocodylians, *A. gomesii*, pennaraptorans, *P. polyodon*, and several ornithischians. Alternatively, the ossified uncinate processes in pennaraptorans could have been modified from a cartilaginous precursor that was widespread in extinct archosaurs, and perhaps even plesiomorphic for Archosauria, but mostly absent in the fossil record because calcification of the cartilage was rare and preservation potential in the absence of calcification was low.

In this contribution, we establish features associated with the attachment of uncinate processes to vertebral ribs as osteological correlates in extant birds and crocodylians. We show that these osteological correlates allow the presence of uncinate processes in fossil archosaurs to be inferred, when uncinate processes are not directly preserved. Using these newly identified osteological correlates, we examine the distribution of uncinate processes in extinct members of Archosauria with a focus on members of Dinosauria. Finally, we reconstruct the ancestral states based on the inferred distribution to explore patterns of uncinate process evolution, and consider the implications for the evolution of ventilation on the line to birds.

## Results and discussion

**Establishing uncinate scars as osteological correlates**. In extant crocodylians and skeletally immature extant birds (Fig. 1c–f), the uncinate processes are anchored to the vertebral ribs by soft connective tissue and can be removed. Removal of an uncinate process from a vertebral rib reveals a rugose area, which is usually slightly concave, on the rib's posterior margin (Fig. 1g–l). We term this rugose area an uncinate scar. In the six skeletally immature extant birds and six extant crocodylians we examined, uncinate scars were consistently present on the vertebral ribs of the anterior thorax. The irregular, rugose surface of an uncinate scar contrasts with the smooth cortex constituting the majority of the vertebral rib's surface and is often perforated by a small number of visible nutrient foramina. The largest uncinate scars, in terms of area, typically occur on the second dorsal vertebral rib and/or the ribs of adjacent vertebral segments.

Typical avian uncinate scars are suboval, heavily rugose, and positioned near the midpoint of the vertebral rib, and tend to taper proximally and/or distally (Fig. 1g, h). Small prominences protrude from the vertebral ribs at the proximal and distal ends of the uncinate scar, contributing to the scar's concavity and potentially causing the scar to appear in lateral view as a slight embayment along the margin of the vertebral rib. Avian uncinate scars vary considerably in width and degree of tapering, both among taxa and depending on anteroposterior position within the thorax. They can be slender and strip-like in extreme cases. However, the avian uncinate scars examined in this study were all heavily rugose.

Unlike in birds, uncinate scars in crocodylians are positioned close to the distal ends of the vertebral ribs. Nevertheless, crocodylians differ from birds in having a third, intermediate segment between each thoracic vertebral rib segment and the corresponding sternal segment[14], and crocodylian uncinate scars are probably positionally equivalent to avian ones if the intermediate ribs of crocodylians are homologous to the distal

parts of the vertebral ribs of birds, as suggested by evidence from embryonic quails[32,33]. Typical uncinate scars in crocodylians are slender strips that range topographically from being distinctly concave as in birds to being essentially flat and inconspicuous. The degree of rugosity is more variable than in avian uncinate scars, and can be quite light. A distal prominence is sometimes present, but we observed no proximal prominences in the specimens we examined. In some cases, the uncinate scar is interrupted along its length by one or two areas of smooth cortex, where the uncinate process was presumably not directly anchored by connective tissue to the vertebral rib. The separate areas of rugosity forming the interrupted scar are always aligned proximodistally and may be situated within a single concavity (Fig. 1j, l). The suboval shape of avian uncinate scars and the elongate shape of crocodylian ones presumably reflect the prong-like and tab-like forms of the uncinate processes seen in birds and crocodylians, respectively.

Uncinate scars nearly identical in morphology and position to those of extant crocodylians were found on two incomplete crocodylian vertebral ribs (AMNH 7900) from the Miocene of Florida, United States (Fig. 1m, n). The precise taxonomic identity of AMNH 7900 is unknown, but the vertebral ribs have the characteristic morphological features seen in extant crocodylians, including the presence of well-developed anterior and posterior intercostal ridges. This observation demonstrates that uncinate scars are sometimes visible in fossil archosaurs, and represent viable osteological correlates of uncinate process attachment in extinct taxa.

**Uncinate scars outside Aves and Crocodylia**. Uncinate scars were found on disarticulated dorsal or posterior most cervical vertebral ribs representing at least 19 fossil members of the avian and crocodylian stem lineages, and on several more archosaur specimens that could be referred only to suprageneric clades (see Table 1 of Supplementary Information for details). Identification of uncinate scars in fossil archosaurs requires that at least the midshaft portion of the vertebral rib be preserved with minimal surficial damage. The rarity of such good preservation probably accounts in part for the fact that relatively few uncinate scars were identified in this study. The observed uncinate scars vary in form, but are generally consistent among closely related taxa (e.g. within cerapodans). These uncinate scars can be distinguished from attachment sites of hypaxial muscles (e.g. mm. intercostales) based on their anatomical positioning, as interpreted in light of the typical distribution of muscle attachment areas on the vertebral ribs of extant archosaurs (see section 2 of Supplementary Information for details). All observed uncinate scars are positioned near the midshafts of vertebral ribs, as in extant birds.

Among theropods examined in this study, the uncinate scars found in the tyrannosaurid *Albertosaurus sarcophagus* (TMP 99.50.41, 99.50.42), the allosaurid *Allosaurus fragilis* (AMNH 5753), and the dromaeosaurid *Saurornitholestes langstoni* (TMP 88.121.39) resemble their avian counterparts. In the tyrannosaurids *Gorgosaurus libratus* (UALVP 10) and *Daspletosaurus torosus* (CMN 8506), along with four indeterminate tyrannosaurid specimens, each uncinate scar is associated with a ridge that lies medially adjacent to the scar's proximal portion, which recedes into the rib shaft in the proximal direction. This feature, which we term the 'proximal ridge', is present in all tyrannosaurids we examined, except *Al. sarcophagus* (TMP 99.50.41, 99.50.42). Uncinate scars in tyrannosaurids (Fig. 2a, b) typically approximate the vertebral ribs in width, and the prominent proximal ridges are clearly visible in lateral view. However, an uncinate scar on a small vertebral rib most definitely belonging to a juvenile

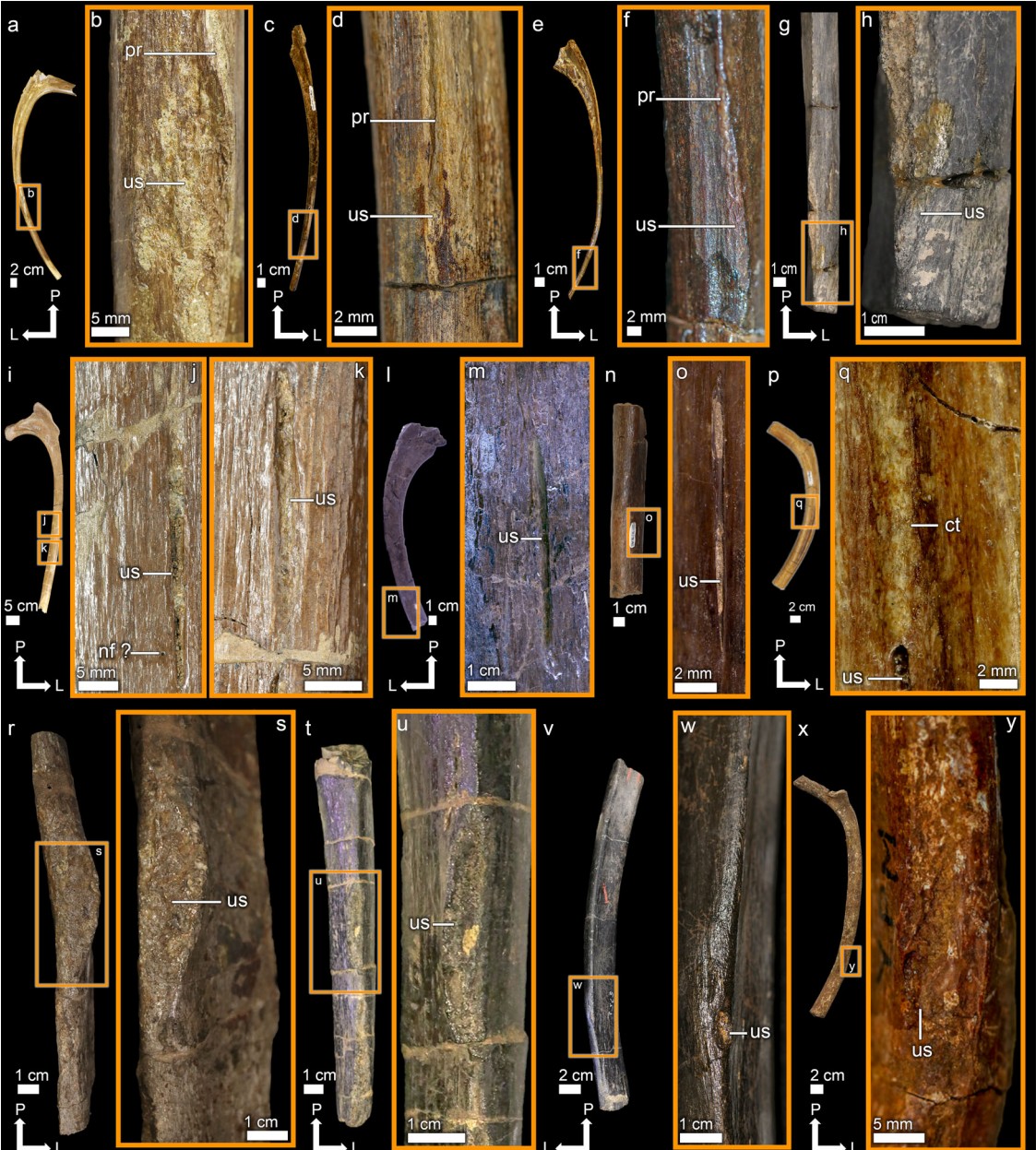

**Fig. 2 Vertebral ribs with uncinate scars in fossil archosaurs.** Left vertebral rib referred to Tyrannosauridae indet. (TMP 92.36.1231) (**a**), and close-up of uncinate scar in posterior view (**b**); right vertebral rib of juvenile individual referred to Tyrannosauridae indet. (TMP 94.12.960) (**c**), and close-up of uncinate scar in posterior view (**d**); right vertebral rib of the ornithomimisaurian theropod *Struthiomimus altus* (AMNH 5355) (**e**), and close-up of uncinate scar in posterior view (**f**); right vertebral rib of the sauropod *Apatosaurus excelsus* (YPM 1981) (**g**), and close-up of uncinate scar in posterior view (**h**); right vertebral rib of the hadrosaur *Gryposaurus notabilis* (AMNH 5350) (**i**), and close-up of uncinate scar in posterior view (**j**, **k**); left vertebral rib of the ceratopsid *Pachyrhinosaurus lakustai* (UALVP 57289) (**l**), and close-up of uncinate scar in posterior view (**m**); vertebral rib referred to ceratopsid *Centrosaurus* sp. (TMP 96.176.135) (**n**), and close-up of uncinate scar in posterior view (**o**); right vertebral rib referred to ceratopsid *Centrosaurus* sp. (TMP 82.19.41) (**p**), and close-up of uncinate scar in posterior view (**q**); vertebral rib of ankylosaur *Sauropelta edwardsi* (AMNH 3032) (**r**), and close-up of uncinate scar in posterior view (**s**); vertebral rib of the stegosaur *Stegosaurus stenops* (AMNH 650) (**t**), and close-up of uncinate scar in posterior view (**u**); left vertebral rib referred to stegosaur *Stegosaurus* sp. (AMNH 5752) (**v**), and close-up of uncinate scar in posterior view (**w**); pseudosuchian right vertebral rib referred to Phytosauria indet. (YPM 6649) in posteromedial view (**x**), and close-up of uncinate scar in posterior view (**y**). ct calcified tissue, L lateral, nf? potential nutrient foramen, P proximal, pr proximal ridge, us uncinate scar.

tyrannosaurid (TMP 94.12.960) (Fig. 2c, d) is a small, suboval concavity, occupying approximately one fifth of the width of the rib shaft, and the proximal ridge is a minor protuberance. The uncinate scars and associated proximal ridges are clearly best developed in mature tyrannosaurids, which suggests that they may have become enlarged in response to muscular loads imposed on the uncinate processes during development. The

uncinate scar and the proximal ridge in the ornithomimisaurian theropod *Struthiomimus altus* (AMNH 5355) (Fig. 2e, f) resemble that of TMP 94.12.960.

Among cerapodans examined in this study, the uncinate scars found in the hadrosaurs *Gryposaurus* (AMNH 5350, 5456) and *Bactrosaurus johnsoni* (AMNH 6553), the non-dryomorph iguanodontian *Tenontosaurus tilletti* (AMNH 3040), the

ceratopsid *Centrosaurus* sp. (TMP 82.18, 96.176.135, ROM 767) and *Pachyrhinosaurus lakustai* (UALVP 57289), the leptoceratopsid *Leptoceratops gracilis* (CMN 8889), and the thescelosaurids *Parksosaurus warreni* (ROM 804) and *Zephyrosaurus schaffi* (MCZ 4392), along with some indeterminate members of Cerapoda resemble their crocodylian counterparts. The cerapodan uncinate scars (Fig. 2i–q) comprise one or two slender strips that are lightly sculpted, and sometimes situated in a concavity (Fig. 2o) as in crocodylians. In *Centrosaurus* sp. (TMP 82.18.41), what appears to be a mass of calcified connective tissues partially covers the uncinate scar externally (Fig. 2q). Though superficially resembling bite marks left by carnivorous vertebrates, the cerapodan uncinate scars are unlike bite marks in their consistent shape, positioning, and orientation, and in that there are never more than two of them on a given vertebral rib (see section 2 of Supplementary Information for details).

The uncinate scars found in the ankylosaurs *Panoplosaurus mirus* (ROM 1215), *Edmontonia longiceps* (CMN 8531), *Sauropelta edwardsi* (AMNH 3032), and *Euoplocephalus tutus* (AMNH 5337), the stegosaurs *Stegosaurus* spp. (AMNH 650, 5752, YPM 1856), (Fig. 2r–w), the sauropod *Apatosaurus excelsus* (YPM 1981) (Fig. 2g, h), an indeterminate aetosaur (NMMNH P50048), and two indeterminate phytosaurs (YPM 6649, NMMNH P60401) (Fig. 2x, y) generally resemble avian uncinate scars. However, the uncinate scars in ankylosaurs each extend onto a shelf that projects posterolaterally from the vertebral rib, enabling the scar to exceed the vertebral rib shaft in width. Those in *Stegosaurus* and *Ap. excelsus*, by contrast, are proportionally narrower than their avian counterparts. Of the two phytosaurs we examined, the uncinate scars found in YPM 6649 resemble avian uncinate scars, whereas the one in NMMNH P60401 is narrow and proximodistally elongated, occupying the majority of the preserved rib's margin. Although the uncinate scar in the examined aetosaur (NMMNH P50048) is incompletely preserved, the preserved portion is wide and tapering as in birds.

**Ancestral state reconstruction for uncinate processes in archosaurs**. An ancestral state reconstruction of the distribution of uncinate processes in archosaurs, based on mapping available data onto an informal consensus cladogram of Archosauria (supertree), suggests that the presence of cartilaginous uncinate processes represents the plesiomorphic condition for both Dinosauria and Archosauria (Fig. 3) (see section 3 and Table 2–5 of Supplementary Information for details). Incorporating branch lengths consistently increased the estimated probability that cartilaginous uncinate processes were ancestral for Archosauria, regardless of the exact approaches used for performing the ancestral state reconstruction and for coding the presence and absence of uncinate processes. Using our preferred coding approach, in which absence of uncinate scars was coded as uncertainty, both maximum parsimony and maximum likelihood consistently recovered the presence of cartilaginous uncinate processes as the most likely condition at Archosauria ($p_{ml} \geq 0.98$) and Dinosauria ($p_{ml} = 1.00$), and the presence of ossified uncinate processes as the most likely condition at Maniraptoriformes ($p_{ml} \geq 0.90$) and Pennaraptora ($p_{ml} \geq 0.99$). Bayesian inference also recovered cartilaginous uncinate processes as the most likely condition at Archosauria ($p_{mb} = 0.65$), but only when branch length estimates were included in the analysis. The ancestral conditions at Dinosauria, Maniraptoriformes, and Pennaraptora could not be recovered with Bayesian inference ($p_{mb} \approx 0.33$) either with or without branch length estimates, which likely reflect the fact that only a small number of archosaurs in the informal consensus cladogram had observed uncinate scars.

Using an alternate coding approach in which uncinate processes were coded as absent in taxa represented by five or more dorsal vertebral ribs that all lacked uncinate scars, nine archosaur taxa were coded as lacking uncinate processes. Under this alternate coding, maximum parsimony recovered results identical to those from the preferred coding. Maximum likelihood and Bayesian inference recovered cartilaginous uncinate processes as the most likely condition at Archosauria ($p_{ml} = 0.61$, $p_{mb} = 0.55$), but only when branch length estimates were incorporated. By contrast, maximum likelihood excluding branch length estimates recovered the absence of uncinate process as the most likely condition at Archosauria ($p_{ml} = 0.92$). The ancestral condition at Dinosauria could not be recovered with confidence using either maximum likelihood or Bayesian inference ($p_{ml} = 0.57$, $p_{mb} \approx 0.33$). Maximum likelihood recovered ossified uncinate processes as the most likely condition at Maniraptoriformes ($p_{ml} = 0.90$) and Pennaraptora ($p_{ml} = 0.99$), but only when branch length estimates were excluded from the analysis. Bayesian inference could not confidently recover the ancestral conditions at Maniraptoriformes ($p_{mb} = 0.33$) and Pennaraptora ($p_{mb} = 0.33$).

As the study sample included only four non-crocodylian pseudosuchian taxa with uncinate scars, the ancestral presence of cartilaginous uncinate processes is less well supported for Archosauria as a whole than for Dinosauria. Including vertebral ribs from more fossil pseudosuchians in future studies would be warranted as a further test of whether uncinate processes were plesiomorphically present in archosaurs.

**Potential homology of uncinate processes across Archosauria**. Uncinate scars in various fossil archosaurs, together with the relatively few known examples of preserved uncinate processes[30,34,35], provide a strong basis for inferring that uncinate processes were widespread and plesiomorphic in Dinosauria. Although only four non-crocodylian pseudosuchians were found to have uncinate scars (two indeterminate phytosaurs, one indeterminate aetosaur, and the notosuchian *A. gomesii*), the relative consistency in morphology and anatomical position of all uncinate scars observed in this study suggests that uncinate processes could well be homologous across Archosauria, as opposed to having evolved independently in multiple lineages. If this is indeed the case, and the correlation in shape between uncinate processes and uncinate scars observed for Aves and Crocodylia holds true outside these groups, then uncinate processes possibly were tab-like in ancestral archosaurs, remained tab-like in pseudosuchians and cerapodans, and became more prong-like in theropods. This scenario is congruent with the presence of preserved tab-like uncinate processes in some ornithischians[30], and would imply that uncinate processes acquired a prong-like morphology before they became ossified structures on the evolutionary line to birds.

Although phytosaurs were treated as pseudosuchians in this study, they were placed outside Archosauria in several phylogenetic studies[36,37]. If the latter is correct, the inferred presence of uncinate scars in phytosaurs would suggest a possible pre-archosaurian origin of uncinate processes, but the wider distribution of these structures across Sauropsida remains to be investigated. Although ossified uncinate processes were recovered as the ancestral state for Maniraptoriformes in our analysis, this result is highly dependent on the presence of ossified uncinate processes in the ornithomimosaurian theropod *P. polyodon*[26], and may reflect the relatively sparse sampling of non-pennaraptoran maniraptoriforms. Thus, it is possible that uncinate ossification evolved separately in *Pelecanimimus* and Pennaraptora.

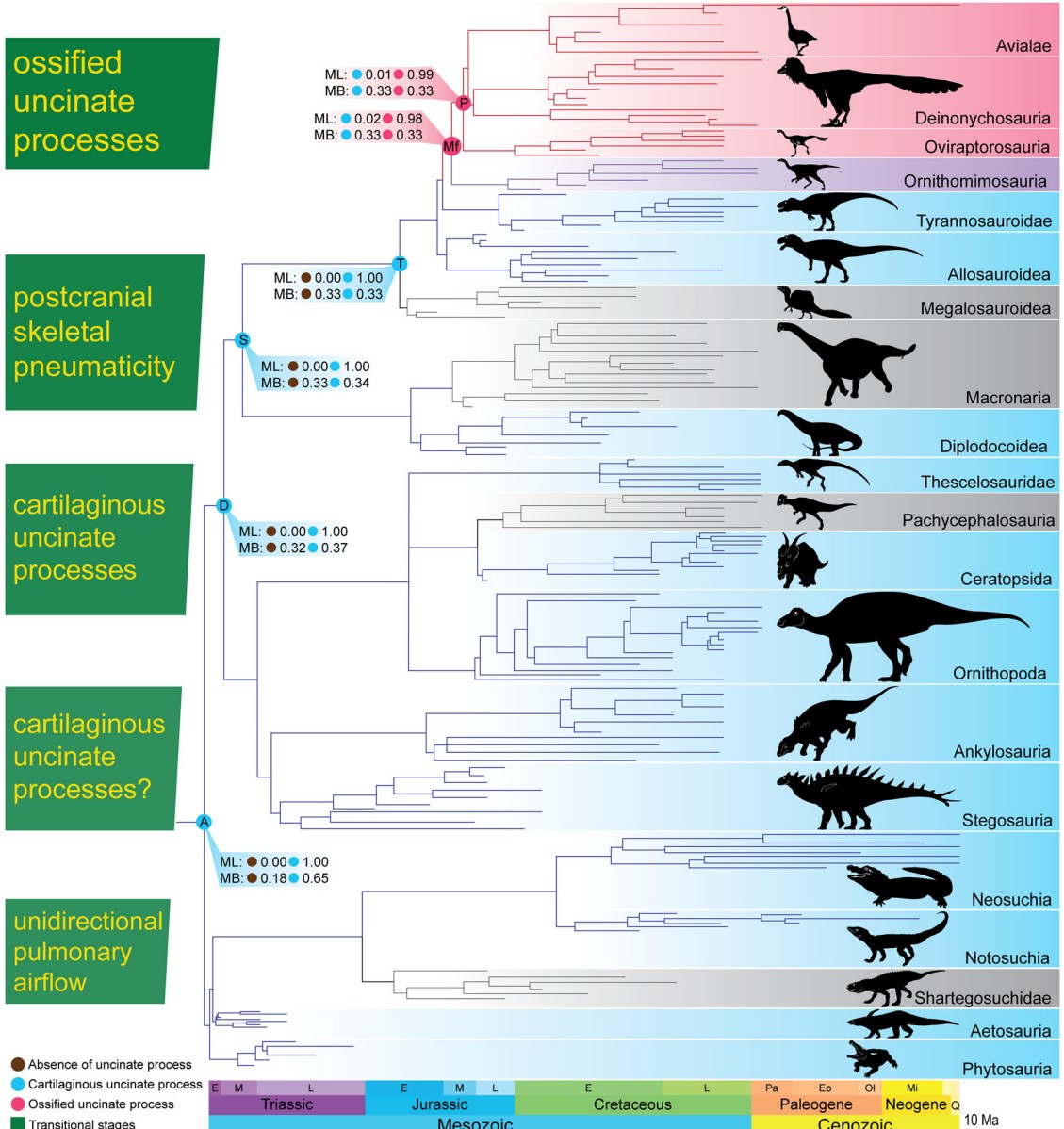

**Fig. 3 Ancestral state reconstruction of uncinate processes in archosaurs.** Results of ancestral state reconstruction on simplified version of informal consensus cladogram with branch length estimates, using preferred coding approach described in this study. Detailed description and raw data are provided in sections 3 to 6 of Supplementary Information. Nodes of interest are coloured according to the results from maximum parsimony, and estimated probabilities from maximum likelihood and Bayesian inference are indicated using text. Major clades of Archosauria with evidence of cartilaginous uncinate processes are labelled and shaded blue; clades with evidence of ossified uncinate processes are labelled and shaded pinkish red; clade with evidence of both cartilaginous and ossified uncinate processes is labelled and shaded purple; and clades for which no evidence is available are labelled and shaded grey. Complete results of ancestral state reconstruction for major clades are given in section 7 of Supplementary Information. Stages of hypothetical evolutionary scenario are outlined to left of cladogram. A Archosauria, D Dinosauria, E Early, Eo Eocene, L Late, M Middle, Mf Maniraptoriformes, Mi Miocene, MB Bayesian inference, ML maximum likelihood, Ol Oligocene, P Pennaraptora, Pa Paleocene, Q Quaternary, T Theropoda.

**Uncinate processes and the evolution of ventilation on the line to birds**. Definitive evidence of parabronchi, and by extension cross-current gas exchange, is lacking from the fossil record, and the origin of high-efficiency gas exchange in birds is difficult to pinpoint. However, the presence of unidirectional pulmonary airflow in crocodylians and lepidosaurs[38], in combination with the widespread distribution of uncinate processes in archosaurs, suggests that components of the avian ventilatory system were likely acquired gradually on the evolutionary path towards birds. Inferences can therefore be made regarding the emergence of individual components of the avian ventilatory system, leading to a hypothetical evolutionary scenario amenable to future testing (Fig. 3). Tab-like, cartilaginous uncinate processes were likely present in ancestral non-avian dinosaurs and potentially present in ancestral archosaurs, as indicated by our ancestral state reconstruction. Such uncinate processes were presumably quite flexible compared to the ossified ones in pennaraptorans, but their advent could nevertheless have resulted in a limited increase in both the mechanical advantage of the ventilatory muscles[15,16] and the structural stability of the trunk[11–13], although whether even the ossified uncinate processes of extant birds have much impact on trunk stability has not yet been rigorously tested. Furthermore, early uncinate processes likely provided additional surface area for ventilatory muscle attachment. By allowing development of larger

muscles, potentially with enhanced moment arms, the uncinate processes may have provided the anatomical capacity to meet higher metabolic demands, because the ventilatory muscles could rotate the vertebral ribs with greater torque to generate inspiratory motions. The potential widespread occurrence of uncinate processes in archosaurs suggests selection pressures may have favoured the enhanced ventilatory performance resulting from the increased volumes of musculature and the mechanical leverage conferred by the uncinate processes. Such an inference would be congruent with the hypothesis that low ambient atmospheric oxygen levels in the Triassic Period selected for adaptations that enhanced ventilatory performance, as well as lower barriers to gas exchange, in ancestral archosaurs[39]. The emergence of postcranial skeletal pneumaticity in saurischians[40] likely also increased metabolic efficiency by replacing heavy, energetically expensive bone with pneumatic space[41,42], allowing energy to be budgeted for other biological demands (e.g. increased activity). Resting metabolic rates in extant birds are positively correlated with uncinate process length[43], suggesting that elevated resting metabolic rates were likely present in fossil maniraptorans with long, ossified uncinate processes[6]. The inferred increase in metabolism in maniraptorans has been related to the origin of powered flight, but this hypothesis cannot be readily tested at present because the inferred metabolic increase and the origin of avian powered flight are difficult to pinpoint in time[44]. However, this uncertainty does not detract from the possibility that uncinate processes were present as an adaptation for improved ventilatory performance in ancestral archosaurs, and were retained and augmented in a wide range of extant and extinct members of the group. Uncinate processes accordingly may have played an important role in ventilation since the dawn of Archosauria.

## Methods

Dorsal or posterior most cervical vertebral ribs of extant and fossil archosaurs housed in museum collections were directly examined to determine whether uncinate scars were present, and by extension whether the presence of uncinate processes could be inferred. Two discrete coding approaches were used in mapping the distribution of uncinate processes across Archosauria and reconstructing ancestral states with respect to this feature. Under our preferred approach, uncinate processes were coded as present in a fossil archosaur if at least one specimen possessed either an uncinate scar or a preserved uncinate process. In the absence of such evidence, the condition was coded as uncertain. Our alternative coding approach, which was used to test the stability of the results obtained under our preferred coding approach, differed in that uncinate processes were coded as absent in taxa for which at least five vertebral ribs were available, regardless of their state of preservation, and showed no sign of uncinate processes or uncinate scars. This resulted in coding uncinate processes as absent in nine taxa. The proterochampsian archosauriform *Chanaresuchus bonapartei* was selected as an outgroup to Archosauria, and uncinate processes were coded as absent for *Ch. bonapartei* because evidence of uncinate processes or uncinate scars was lacking from a total of 13 well-preserved vertebral ribs observed in two individuals (MCZ 4037, 4038). The ancestral state reconstruction (ASR) was performed using RStudio 4.1.2, using an informal species-level cladogram (supertree) compiled from the results of 23 phylogenetic studies of varying scope across Archosauria[45–63]. To add a phylogenetic tree to the informal cladogram, at least one common taxon present in both trees was used as a topological landmark, and the phylogenetic tree was grafted onto the informal cladogram at the phylogenetic position of the common taxon. Branch lengths were estimated based on first and last appearance data[64]. Maximum parsimony, maximum likelihood, and Bayesian inference were used to infer ancestral states. Each of these methods was applied using both our preferred and alternate coding approaches, and both with and without branch length estimates. Detailed procedures, R script, and raw data for the ASR are provided in the Supplementary Data.

**Reporting summary**. Further information on research design is available in the Nature Portfolio Reporting Summary linked to this article.

## Data availability

Data supporting the findings of this study include (1) a list of specimens examined and (2) raw data and R script to perform the ancestral state reconstruction are provided in the Supplementary Data. Additional images of uncinate processes and uncinate scars are available from the corresponding author upon request.

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

## Acknowledgements

This research was funded by the Natural Sciences and Engineering Research Council of Canada (Discovery Grant RGPIN-2017-06246) and start-up funding awarded by the University of Alberta to C.S., National Foundation Science Grant (DBI 0743327) awarded to L.P.A.M.C., and a Student Research Project from the Dinosaur Research Institute awarded to Y.Y.W. We are grateful to Carl Mehling (AMNH), Binghe Di (IVPP), Eva Biedron (MCZ), Nicole Ridgwell (NMMNH), Braden Barr (UAMZ), Brandon Strilisky (TMP), Clive Coy, Howard Gibbons, and Robin Sissons (UALVP), and Daniel Brinkman (YPM) for access to archosaur specimens. We thank Oksana V. Vernygora and Nicolàs E. Campione for technical advice on performing the ancestral state reconstruction using R. We thank members of the Sullivan, Currie, and Caldwell labs at the University of Alberta for discussion. We thank Sydney Mohr and Khoi Nguyen for the permissions to adapt their artworks into silhouettes. We thank Brad McFeeters, Craig Dylke, FunkMonk, Jagged Fang Design, Mariana Ruiz, Michael Keesey, Raven Amos, Steven Traver, and Tasman Dixon for uploading their artwork to the public domain section of the platform PhyloPic.

## Author contributions

Y.Y.W. and C.S. designed the project; Y.Y.W. collected data from extant and fossil specimens; Y.Y.W., L.P.A.M.C. and C.S. analysed and interpreted the data; Y.Y.W., L.P.A.M.C. and C.S. wrote and reviewed the manuscript.

## Competing interests

The authors declare no competing interests.

## Ethics/ethical approval

This study is a collaboration among a Canadian professor, a Dutch professor, and a Chinese student they supervise. All authors contributed equally to the findings of this study. The authors declare no competing interests.
