## [Peer Review File · Communications Biology]

Reviewers' comments:

Reviewer #1 (Remarks to the Author):

Dear Wang, Claessens and Sullivan (Hi Yan-Yin, Leon and Corwin!),

Thank you for the opportunity to review your work. I enjoyed reading your article. It is nice to know that an osteological correlate like the one you proposed is available. I thought your data sampling among dinosaurs was good but I only counted two non-crocodylian pseudosuchians that you sampled. It would be great to see more of the latter as this will make your reconstruction of the Archosaur ancestral condition more robust. As is, I recommend to narrow your scope to dinosaurs and modern crocodylians as I think this would still make an interesting paper suited for this journal. I hope you find my comments to be helpful. Please feel free to reach out if you have any questions.

I look forward to seeing this out.

Best regards,

Michael Pittman
The Chinese University of Hong Kong

Reviewer #2 (Remarks to the Author):

This is a well-written and rigorous paper that brings significant new data to an interesting topic: breathing mechanisms in modern and fossil archosaurs. The paper focuses on uncinat processes - bony or cartilaginous attachments to the ribs that appear to be important in lung ventilation in modern archosaurs - and demonstrates for the first time an osteological correlate of these processes that allows them to be identified in fossil taxa where the processes may not have been ossified. The distribution of this osteological correlate in fossil taxa is then used to argue that uncinat processes are ancestral for Archosauria.

I am fully supportive of publication with minimal revisions. I note only the following minor points.

- 1) Various taxonomic names are mentioned at various points without explaining them. For the benefit of the reader I would suggest "anhimid birds" rather than "anhimids", "neognath birds" instead of "neognaths" etc.
- 2) At more than one point it is asserted that uncinat processes cannot be ancestral for saurians because they are absent in squamates (lizards and snakes). But if they are absent in squamates but present in rhynchocephalians, then the ancestral condition for Lepidosauromorpha is ambiguous, as is the ancestral condition for Sauria. You should tweak your text to reflect this ambiguity.
- 3) On line 69 thescelosaurids and ornithopods are listed as if they are different taxa. But thescelosaurids are ornithopods.
- 4) A Miocene crocodylian specimen AMNH 7900 is referred to on line 117 as an indeterminate croc. But in figure 1 the same specimen is referred to as Alligator mississippiensis (and the caption does not make clear that this is a Miocene specimen).
- 5) Capitalise Cerapoda on line 198
- 6) In Figure 3, the grey clades are missing data, but on the figure itself these are referred to as "absence of uncinat". These are quite different things. Change the figure wording to reflect that this

is missing data not genuine absence.

Reviewer #3 (Remarks to the Author):

a really interesting paper focussing on an important topic - comments and suggestions below:

line 44: are there any empirical data that support the mechanical hypothesis as both of these are text book references - do the authors think they should present this as a viable alternate hypothesis?

line 73 - could a reference to the presence of cartilaginous uncinata in crocodylians be added to flesh out this statement

line 215 - wouldn't this suggest that these were present in the ancestor of both crocodylians and birds ?

Point by point responses to reviewers' comments for Deep reptilian evolutionary roots of a major avian respiratory adaptation

Yan-yin Wang¹, Leon P.A.M. Claessens², Corwin Sullivan^{1,3}

¹Department of Biological Sciences, CW 405 Biological Sciences Building, University of Alberta, Edmonton, AB, T6G 2E9, Canada

²Maastricht Science Programme, Faculty of Science and Engineering, Maastricht University, Maastricht, The Netherlands

³Philip J. Currie Dinosaur Museum, Wembley, AB, T0H 3S0, Canada

Corresponding Author: Yan-yin Wang

Email: yanyin@ualberta.ca

In this document, line numbers in purple refer to the lines of the manuscript that the reviewers commented on. Line numbers in the revised change section refer to the lines of the revised manuscript submitted in this package.

Reviewer 1 (Dr Michael Pittman)

1. Line 29: The fossil sample dataset has a good number of fossil taxa but does not include pterosaurs to represent non-dinosaurian ornithodirans and has two taxa to cover non-crocodylian pseudosuchians (a notosuchian and phytosaurs by my count). The dataset is best suited to commenting on the Dinosauria ancestral condition and an edited paper with this narrowed focus would fit the requirements of this journal. As the manuscript, I believe non-crocodylian pseudosuchians from the major subclades need to be sampled to more robustly reconstruct the archosaur ancestral condition.

Revised changes:

Araripesuchus and three non-crocodylian pseudosuchians (NMMNH P50048, P60401, and YPM 6649) are the four non-crocodylian pseudosuchian taxa with uncinata scars sampled in this study. Sentences in the section “Ancestral state reconstruction for uncinata processes in archosaurs” and “Potential homology of uncinata processes across Archosauria” have been modified, so that the number of non-crocodylian pseudosuchians are explicitly mentioned.

The results at the node Archosauria have been retained because they suggest the possibility that uncinata processes are homologous across Archosauria. Although this result is tentative, including it in the paper indicates to the reader where our knowledge of the origin of uncinata processes stands at present, and will hopefully encourage further sampling of uncinata scars in fossil pseudosuchians.

However, sentences are added to line 197, to acknowledge that additional sampling from fossil pseudosuchians would be warranted in order to bolster the current result for Archosauria. The discussion in the section “Potential homology of uncinata processes across Archosauria” is

modified in tone, to reflect the fact that few non-crocodylian pseudosuchians were sampled in this study.

2. Line 35: Any supporting references you could add?

Revised changes: Line 34. Reference has been added for the absence of uncinat processes in anhimid and megapodid birds, and for the fact that avian uncinat processes are normally ossified.

Line 41. Reference has been added for the fusion of uncinat processes to the vertebral ribs in birds. The presence of uncinat processes on the posteriormost cervical ribs, and the fusion between uncinat processes and their corresponding vertebral ribs observed in the avian skeletal materials housed at the University of Alberta Museum of Zoology (e.g. *Falco sparverius* UAMZ 4022) has been retained to support the referred literature.

3. Line 42: Any supporting references you could add?

Revised changes: Line 41. Osteology of the Reptiles by Romer (1956) has been added as a supporting reference.

4. Line 44: It would be helpful to explicitly state that the hypotheses are not mutually exclusive to help the non-specialist reader.

Revised changes: Line 42. sentence has been changed to “Two main hypotheses for the function of avian uncinat processes, which are not mutually exclusive have been proposed: mechanical reinforcement of the ribcage and ventilation”.

5. Line 66: Recommend to use ‘saurians’ consistently, e.g., stegosaurians. Or you could just replace the use of ‘saurians’ in some places to ‘saurs’

Revised changes: ‘-saurian(s)’ has been changed to ‘-saur(s)’ throughout the text, except where a taxonomic name is being used as a modifier (lines 43, 64, and 125).

6. Line 69: This datapoint should be included in the main tree figure as non-crocodylian pseudosuchians are currently not represented there. The fact that the uncinat processes are calcified in this taxon suggests that sampling other non-crocodylian pseudosuchians would be important for constraining the Archosaurian ancestral condition.

Revised changes: Line 551. Figure 3 has been modified to include Notosuchia, in which *Araripesuchus* is placed, Aetosauria in which Aetosauria indet. (NMMNH P50048) is placed, and Phytosauria in which the sampled Phytosauria indet. (YPM 6649, NMMNH P60401) is placed.

Shartegosuchidae has been added to indicate that not all pseudosuchian lineages have been sampled in this study.

7. Line 84: There are only a few so I recommend to just say ‘select notosuchians and ornithischians’.

Revised changes: Line 67. The sentence has been changed to “However, the selected notosuchian and ornithischian examples listed above notwithstanding, preserved uncinat processes are rarely found outside Pennaraptora.”.

8. Line 136: It would be very helpful to add a figure part showing what the cartilaginous part looks like as some readers might not have seen it before.

Revised changes: Line 502. New panels have been added to Fig 1 to illustrate (1) ossified uncinat process fused to vertebral rib, (2) ossified uncinat process unfused to vertebral rib, and (3) cartilaginous uncinat processes.

8. Line 136: Recommend to include clades before the genera as some readers won’t be familiar with theropod phylogeny e.g., the dromaeosaurid *Saurornitholestes langstoni* (TMP 88.121.39).

9. Line 150: Recommend to include clades before the genera as some readers won’t be familiar with ornithischian phylogeny e.g., in the hadrosaurs *Gryposaurus* (AMNH 5350, 5456) and *Bactrosaurus johnsoni* (AMNH 6553), ...

10. Line 162: Recommend to include clades before the genera as some readers won’t be familiar with ornithischian and sauropod phylogeny e.g., diplodocid sauropod *Apatosaurus*

11. Line 441: Adding clade names will be very helpful for reader that are not familiar with bird or crocodylian taxonomy.

12. Line 453: Same comment about clade names to help non-specialist readers.

Revised changes: Clade names have been added before the genera throughout the main text.

13. Line 165: This additional non-crocodylian pseudosuchian data point should be added to the main tree figure to better convey your taxon sampling.

Revised changes: Line 551. Figure 3 has been modified to include *Phytosauria* and *Aetosauria*, which contain the sampled specimens (i.e. *Phytosauria* indet. and *Aetosauria* indet.).

14. Line 201: You could mention your notosuchian datapoint here. It seems it and the phytosaur (if it is an archosaur) are the only non-crocodylian pseudosuchian taxa in your dataset?

Revised changes: Line 205. *Araripesuchus* and three non-crocodylian pseudosuchians (NMMNH P50048 and P60401, and YPM 6649) are the four non-crocodylian pseudosuchians with uncinata scars sampled in this study.

The sentence has been changed to “Although phytosaurs were positioned as basal pseudosuchians in this study, along with the indeterminate aetosaur and the notosuchian *A. gomesii*, they have been recovered outside Archosauria in several recent phylogenetic studies.”.

15. Line 205: Can you spell out what you mean? Are the ribs smooth with no scars?

Revised changes: Line 219. The sentence has been changed to state that “...the wider distribution of these structures [i.e. uncinata processes] across Sauropsida remains to be investigated”, to clarify that we have not examined lepidosauromorphs in this study.

16. Line 210: Are there any implications for the hypothesis about the mechanical reinforcement of the ribcage? If not, it is worth stating this anyway since you went into detail about it in the introduction.

Revised changes: Line 235. Some text has been added to suggest that the advent of uncinata processes may have provided some increased structural stability to the trunk, although whether uncinata processes act this way even in modern birds still awaits rigorous analysis. We also have added sentences to mention that the uncinata processes would have provided additional surface area for ventilatory muscle attachment.

17. Line 210: An additional figure showing this across the tree would be very helpful to articulate where we are at now.

Revised changes: Line 551. A panel has been added to illustrate stages of the hypothetical evolutionary scenario described in the paper. We have chosen to add a panel instead an additional figure, because the hypothetical evolutionary scenario is probably clearer if we position the scenario next to the cladogram with results of ancestral state reconstruction.

18. Line 235: Can you be more explicit about why this is?

Revised changes: Line 258. The sentence has been changed to “An increase in metabolism in maniraptorans may be related to the origin of powered flight, which could not be explored further at present because the precise timings of an increased metabolism and the origin of avian powered flight remain uncertain”, to clarify that additional data is warranted for further inference on the evolutionary relationship between the potential increased metabolism and the origin of powered flight.

Line 255. Sentence is added to describe the possibility that resting metabolic rates may be correlated to dimensions of cartilaginous uncinata processes. Accordingly, the resting metabolic rates may have increased before uncinata processes became ossified in pennaraptorans, if uncinata processes became elongated before the origin of pennaraptorans.

19. Line 469: Some of the insets are hard to make sense of. Recommend some further image post-processing to improve contrast etc.

Revised changes: Lines 502 and 527. Font sizes and shape sizes have been adjusted for visibility. Photos have been sharpened to increase contrast where necessary, and colour of the rectangles have changed to orange for better visibility.

20. Line 481: I recommend to show results for all three mapping methods and at more internal nodes. There is a gap between the values visible at the three nodes and the colour coding of the clades that makes it difficult to evaluate the study results. Perhaps colour the branches so the reader can visually see the data within each colour coded clade? I feel that this figure should be self-contained without the need to refer to the supplement or raw data.

The coloured circles at nodes and the rib cartoon were a bit confusing. Recommend to put the cartoon in the key or have the rib at the nodal symbol.

Recommend to make the square key symbol a circle too since the ML and MB values are also referring to nodal values.

Revised changes: Line 551. Figure 3 has been modified to include additional internal nodes for Saurischia and Theropoda (Table 5 in the supplementary information is also updated), to demonstrate the results of ancestral state reconstruction leading to birds.

Internal nodes with values of ancestral state reconstruction are labelled by the initials of clades, to visually indicate their positions better visually on the cladogram. Branches with known uncinata processes or uncinata scars are colour coded and shaded based on the results using maximal parsimony on a time calibrated tree. Probability labels are changed to circles, and rib diagrams are removed.

Mapping all results together is great for comparisons. However, the results of three mapping methods are not identical for all nodes and branches, so mapping all three methods on top of each other would be confusing for readers.

Reviewer 2

1. Various taxonomic names are mentioned at various points without explaining them. For the benefit of the reader I would suggest "anhimid birds" rather than "anhimids", "neognath birds" instead of "neognaths" etc.

Revised changes: Explanatory terms (e.g. birds) have been added after the taxonomic names where appropriate.

2. At more than one point it is asserted that uncinat processes cannot be ancestral for saurians because they are absent in squamates (lizards and snakes). But if they are absent in squamates but present in rhynchocephalians, then the ancestral condition for Lepidosauromorpha is ambiguous, as is the ancestral condition for Sauria. You should tweak your text to reflect this ambiguity.

Revised changes: Line 39, 219. Sentences have been modified to reflect the ambiguity of the ancestral condition in both Lepidosauromorpha and Sauria.

Line 39. Sentence has been changed to “The rod-like cartilaginous uncinat processes of the rhynchocephalian *Sphenodon punctatus* may not be homologous to those of birds and crocodylians: most extant lepidosauromorphs lack uncinat processes.”. Romer ¹ has been added as a reference to support the statement that most extant lepidosauromorphs lack uncinat processes.

Line 219. Sentence has been changed to “... but the wider distribution of these structures across Sauropsida remains to be investigated”.

3. On line 69 thescelosaurids and ornithopods are listed as if they are different taxa. But thescelosaurids are ornithopods.

Revised changes: Madzia, et al. ², one of the phylogenetic studies we consulted in constructing the tree used for the ancestral state reconstruction, recovered thescelosaurids as a group of basal neornithischians outside Ornithopoda. Similar results were obtained in several other studies of ornithischian systematics (e.g. Boyd ³). We therefore have continued to treat thescelosaurids as distinct from ornithopods.

We acknowledge, however, that some authors (e.g. Eberth, et al. ⁴.) indeed continue to regard thescelosaurids as ornithopods, a view that was once widely held.

4. A Miocene crocodylian specimen AMNH 7900 is referred to on line 117 as an indeterminate croc. But in figure 1 the same specimen is referred to as *Alligator mississippiensis* (and the caption does not make clear that this is a Miocene specimen).

Revised changes: Line 117. The specimen is labelled *Alligator mississippiensis* in the American Museum of Natural History collocation as AMNH 7900. However, AMNH 7900 is not associated

with materials that can be decisively assigned to *A. mississippiensis*. We therefore have taken a conservative approach and refer to AMNH 7900 as an indeterminate crocodylian.

The figure caption (line 497) has been changed to read “incomplete right dorsal rib of a taxonomically indeterminate crocodylian from the Miocene of Florida, USA (AMNH 7900) in posteromedial view”.

5. Capitalise Cerapoda on line 198

Revised changes: Line 211. The sentence has been changed to “...remained tab-like in pseudosuchians and cerapodans, and became more prong-like in theropods.”, to maintain consistency in the terminology used for particular clades.

6. In Figure 3, the grey clades are missing data, but on the figure itself these are referred to as "absence of uncinata". These are quite different things. Change the figure wording to reflect that this is missing data not genuine absence.

Revised changes: The grey colour is intended to indicate missing data, and the black colour originally used in the legend was intended to denote the level of probability that uncinata processes were absent at a particular node.

Line 551. The colour used to denote probability of the absence of uncinata process has been changed to caramel brown, to avoid confusion.

Reviewer 3

1. line 44: are there any empirical data that support the mechanical hypothesis as both of these are text book references - do the authors think they should present this as a viable alternate hypothesis?

Revised changes: Line 235. Some text has been added to suggest that the advent of uncinata processes may have provided some increased structural stability to the trunk, although whether uncinata processes act this way even in modern birds still awaits rigorous analysis. We also now mention that the uncinata processes would have provided additional surface area for ventilatory muscle attachment.

2. line 73 - could a reference to the presence of cartilaginous uncinata in crocodylians be added to flesh out this statement

Revised changes: Line 39. Cong, et al. ⁵ has been added as a reference for the presence of cartilaginous uncinata processes in crocodylians.

3. line 215 - wouldn't this suggest that these were present in the ancestor of both crocodylians and birds ?

Revised changes: Unidirectional pulmonary airflow and cartilaginous uncinata processes are considered present in the ancestral archosaurs according to Farmer ⁶ and our ancestral state reconstruction.

Line 228 The sentence has been changed to “However, the presence of unidirectional pulmonary airflow in crocodylians and lepidosaurs, in combination with the widespread distribution of uncinata processes in archosaurs, suggests that components of the avian ventilatory system were likely acquired gradually on the evolutionary path towards birds.”, to state that components of avian ventilatory systems observable in extant birds are likely acquired gradually since the early days of archosaur evolution.

References

- 1 Romer, A. S. The axial skeleton in *The osteology of the reptiles* 275-279 (University of Chicago Press, 1956)
- 2 Madzia, D., Boyd, C. A. & Mazuch, M. A basal ornithomimid dinosaur from the Cenomanian of the Czech Republic. *J. Syst. Palaeontol.* **16**, 967-979; 10.1080/14772019.2017.1371258 (2018).
- 3 Boyd, C. A. The systematic relationships and biogeographic history of ornithomimid dinosaurs. *PeerJ* **3**, e1523; 10.7717/peerj.1523 (2015).

- 4 Eberth, D. A. *et al.* Dinosaur biostratigraphy of the Edmonton Group (Upper Cretaceous), Alberta, Canada: Evidence for climate influence. *Can. J. Earth Sci.* **50**, 701-726; 10.1139/cjes-2012-0185 (2013).
- 5 Cong, L. Y., Hou, L. H. & Wu, X. C. The gross anatomy of *Alligator sinensis* fauvel: Integument, osteology, and myology (in Chinese with English summary) 1 - 388 (China Science Publishing & Media Ltd., 1988)
- 6 Farmer, C. G. Pulmonary transformations of vertebrates in *The biology of the avian respiratory system* 99-112 (Springer, 2017)